EMBO
Molecular Medicine

# Profiling MHC II immunopeptidome of blood-stage malaria reveals that cDC1 control the functionality of parasite-specific CD4 T cells

Marion Draheim[1], Myriam F Wlodarczyk[1], Karine Crozat[2], Jean-Michel Saliou[3,4], Tchilabalo Dilezitoko Alayi[3,4] [ID], Stanislas Tomavo[3,4], Ali Hassan[1], Anna Salvioni[1], Claudia Demarta-Gatsi[5], John Sidney[6], Alessandro Sette[6], Marc Dalod[2], Antoine Berry[1], Olivier Silvie[7] [ID] & Nicolas Blanchard[1,*] [ID]

## Abstract

In malaria, CD4 Th1 and T follicular helper ($T_{FH}$) cells are important for controlling parasite growth, but Th1 cells also contribute to immunopathology. Moreover, various regulatory CD4 T-cell subsets are critical to hamper pathology. Yet the antigen-presenting cells controlling Th functionality, as well as the antigens recognized by CD4 T cells, are largely unknown. Here, we characterize the MHC II immunopeptidome presented by DC during blood-stage malaria in mice. We establish the immunodominance hierarchy of 14 MHC II ligands derived from conserved parasite proteins. Immunodominance is shaped differently whether blood stage is preceded or not by liver stage, but the same ETRAMP-specific dominant response develops in both contexts. In naïve mice and at the onset of cerebral malaria, CD8α[+] dendritic cells (cDC1) are superior to other DC subsets for MHC II presentation of the ETRAMP epitope. Using *in vivo* depletion of cDC1, we show that cDC1 promote parasite-specific Th1 cells and inhibit the development of IL-10[+] CD4 T cells. This work profiles the *P. berghei* blood-stage MHC II immunopeptidome, highlights the potency of cDC1 to present malaria antigens on MHC II, and reveals a major role for cDC1 in regulating malaria-specific CD4 T-cell responses.

**Keywords** CD4 T cell; dendritic cell; malaria; MHC II presentation; *Plasmodium berghei*

**Subject Categories** Immunology; Microbiology, Virology & Host Pathogen Interaction

## Introduction

Malaria is caused by parasites of the *Plasmodium* genus. This disease continues to threaten nearly half of the world's population and to kill more than 400,000 people yearly. Malaria infection leads to a broad spectrum of diseases with varying severity. While some asymptomatic parasite carriers show no clinical signs, individuals with uncomplicated malaria present mild symptoms, like fever and/or myalgia, and severe malaria patients face deadly manifestations, such as anemia or cerebral malaria. The diversity of human malaria pathophysiology can be recapitulated in part using different combinations of mouse backgrounds and rodent-adapted *Plasmodium* species. Altogether, rodent studies have revealed the complex and dual roles of T cells, which seem to be involved both in protection and in pathogenesis (Freitas do Rosario & Langhorne, 2012; Howland *et al*, 2015a).

CD8 T cells are essential to contain parasite development during the initial liver stages (Frevert & Krzych, 2015; Radtke *et al*, 2015b). During blood stage, when parasites reside exclusively within erythrocytes, CD8 T cells may target MHC I-positive parasitized erythroblasts (Imai *et al*, 2013) and be involved in parasite clearance (Safeukui *et al*, 2015). Besides these protective functions, CD8 T cells play a well-established deleterious role in the vascular pathology associated with experimental cerebral malaria (ECM). A hallmark of ECM is the cross-presentation of parasite antigens by endothelial cells of cerebral microcapillaries to CD8 T cells (Howland *et al*, 2015b; Swanson *et al*, 2016). Combined to a restriction in venous blood efflux due to cell sequestration (Nacer *et al*, 2014), the CD8-mediated processes are considered pivotal in the vascular breakdown and fatal condition.

1  Centre de Physiopathologie Toulouse Purpan (CPTP), INSERM, CNRS, Université de Toulouse, UPS, Toulouse, France
2  CNRS, INSERM, CIML, Aix Marseille Université, Marseille, France
3  Centre d'Infection et d'Immunité de Lille (CIIL), CNRS UMR 8204, Inserm U1019, CHU Lille, Institut Pasteur de Lille, University of Lille, Lille, France
4  Plateforme de Protéomique et Peptides Modifiés (P3M), CNRS, Institut Pasteur de Lille, University of Lille, Lille, France
5  CNRS, INSERM, Institut Pasteur, Unité de Biologie des Interactions Hôte Parasites, Paris, France
6  La Jolla Institute of Allergy and Immunology, San Diego, CA, USA
7  INSERM, CNRS, Centre d'Immunologie et des Maladies Infectieuses, Sorbonne Universités, UPMC University of Paris 06, Paris, France
   *Corresponding author. Tel: +33 562 74 83 07; E-mail: nicolas.blanchard@inserm.fr

Being major producers of inflammatory and regulatory cytokines, CD4 T cells are critical fine-tuners of the balance between protection and pathology. On the one hand, CD4 T cells contribute to parasite control. Effector memory Th1 CD4 T cells confer partial protection in the self-resolutive *Plasmodium chabaudi chabaudi* (*Pcc*) model (Stephens & Langhorne, 2010), and loss of T-bet, a master regulator of Th1 differentiation, impairs control of parasitemia in the *Plasmodium berghei* ANKA (*PbA*) model (Oakley *et al*, 2013). Moreover, a subset of CD4 T cells, called T follicular helper (T$_{FH}$) cells, releases the B-cell-helping IL-21 cytokine (Perez-Mazliah *et al*, 2015) and is key to promote effective germinal center formation and anti-parasite humoral immunity (Sebina *et al*, 2016; Zander *et al*, 2016).

On the other hand, CD4 T cells also contribute to malaria-associated pathology. By secreting IFNγ in the first days of infection (Belnoue *et al*, 2002; Villegas-Mendez *et al*, 2012), CD4 T cells promote CXCL9- and CXCL10-dependent CXCR3-mediated accumulation of CD8 T cells in the brain (Campanella *et al*, 2008; Van den Steen *et al*, 2008). Accordingly, T-bet-deficient (Oakley *et al*, 2013) and IL-12Rbeta2-deficient (Fauconnier *et al*, 2012) mice are less susceptible to ECM development. In addition, production of regulatory cytokines, such as IL-10 (Couper *et al*, 2008; Freitas do Rosario *et al*, 2012; Villegas-Mendez *et al*, 2016; Claser *et al*, 2017) and IL-27 (Kimura *et al*, 2016), by CD4 T cells is instrumental for tilting the balance away from pathology. Yet despite the central role of CD4 T cells in regulating protective *versus* deleterious immunity during malaria, their cognate antigens, as well as the antigen-presenting cells (APC) controlling their differentiation, are poorly characterized.

Chief among these APC are the dendritic cells (DC). DC concomitantly act as innate sensors of pathogen motifs, activators of innate immune cells, and initiators of the adaptive T-cell-mediated immunity. DC comprise two major branches: the plasmacytoid DC (pDC) and the conventional DC (cDC), which, based on ontogeny, can be further subdivided into cDC1 and cDC2 (Guilliams *et al*, 2014). Schematically, pDC are viewed as specialized to respond to viral infection by producing type I IFN, and cDC are considered as the most potent T-cell-activating cells (Durai & Murphy, 2016), even though depending on the context, tissue, or species analyzed, these functions may be completely or partially shared.

Early evidence suggested a functional dichotomy in antigen presentation among cDC, with cDC1 being best-equipped for cross-priming CD8 T cells and cDC2 preferentially activating CD4 T cells. This specialization was proposed after specifically targeting the same antigen to each subset (Dudziak *et al*, 2007). Since then, the specific requirement for cDC1 in cross-priming CD8 T cells has been confirmed in viral (Hildner *et al*, 2008; Helft *et al*, 2012), bacterial (Yamazaki *et al*, 2013), and parasite (Mashayekhi *et al*, 2011; Piva *et al*, 2012; Lau *et al*, 2014; Radtke *et al*, 2015a) infections. Furthermore, cDC1 are instrumental to reactivate memory CD8 T cells in response to *Listeria* or virus assaults (Alexandre *et al*, 2016). With respect to CD4 T-cell activation, the respective performances of cDC1 and cDC2 seem to vary according to the context and source of antigen. For yeast-associated antigens, cDC2 perform better than cDC1 (Backer *et al*, 2008) while for cell-associated antigens, cDC1 are more potent than cDC2 (Schnorrer *et al*, 2006). The bacterial antigen presentation capacity of cDC1 and cDC2 appears equivalent when incubated with fixed bacteria (Schnorrer *et al*, 2006). *In vivo*, some studies showed only a limited role for cDC1 in priming CD4 T cells against a soluble antigen (Yamazaki *et al*, 2013) or the West Nile Virus (Hildner *et al*, 2008). In agreement, cDC2 but not cDC1 are necessary for proper T$_{FH}$ differentiation after immunization with allogeneic red blood cell (RBC) (Calabro *et al*, 2016). In contrast, in other contexts such as during *Leishmania major* (Ashok *et al*, 2014; Martinez-Lopez *et al*, 2015) and *Toxoplasma gondii* (Mashayekhi *et al*, 2011) parasite infections, cDC1 are critical to drive Th1 responses. Of note, it has been shown in the course of a viral infection that by interacting with pre-activated CD4 T cells, cDC1 constitute a platform for delivery of CD4 T cell help to CD8 T cells (Eickhoff *et al*, 2015; Hor *et al*, 2015).

In malaria, the respective contributions of DC subsets in controlling CD4 T-cell activation are ill-defined. In the self-resolutive *Pcc* model, cDC2 are more potent than cDC1 for MHC II presentation of two MSP1 epitopes until day 11 post-infection, but for reasons that were not elucidated, the trend is reversed a few days later (Sponaas *et al*, 2006). In this model, a more recent study identified a role for inflammatory monocytes in promoting Th1 responses (Lonnberg *et al*, 2017). During *PbA* infection, MHC II presentation by splenic cDC1 at day 3 was found to be more efficient compared to cDC2 in one study (Lundie *et al*, 2010) but roughly similar to cDC2 in another (Lundie *et al*, 2008). Importantly, this was evaluated in BALB/c mice, which are more resistant to neuroinflammation than C57BL/6 (B6) mice (Hafalla *et al*, 2012), and with a model antigen. In conclusion, the implication of DC subsets in controlling endogenous CD4 T-cell responses during severe malaria remains unsettled.

Here, we profiled the MHC II immunopeptidome presented by DC to CD4 T cells during *Pb* infection and we engineered reporter CD4 T-cell hybridomas specific for the most prominent ETRAMP10.2 epitope. We report that in naïve and malaria-infected mice, cDC1 are more potent than cDC2 for presenting *Plasmodium* antigens and that selective *in vivo* ablation of cDC1 blunts the development of parasite-specific Th1 responses.

## Results

### Profiling the *Plasmodium berghei*-derived MHC II immunopeptidome

Characterizing MHC II ligands and creating parasite-specific reporter T cells are critical steps to understand the modalities of antigen presentation and CD4 T-cell polarization during blood-stage malaria.

We used mass spectrometry to globally characterize *Pb* antigenic peptides presented by MHC II on the surface of DC (Fig 1A). We immunoprecipitated the MHC II molecules from a splenic DC tumor cell line called MutuDC (Fuertes Marraco *et al*, 2012), incubated with *Pb*-parasitized RBC (pRBC). Peptide-loaded MHC II are trimers of I-A$^b$α chain, I-A$^b$β chain, and antigenic peptide, which, depending on the affinity of the loaded peptide, can remain stable in SDS without prior boiling (Natarajan *et al*, 1999). To ensure that our protocol allowed the pull-down of peptide-MHC II trimers and not just of single I-A$^b$β chain, we compared the migration of eluates with and without prior boiling, confirming that the immunoprecipitated material was SDS-stable (Fig 1B). Although genetically very close to *PbA* (Otto *et al*, 2014), several variants of *Pb* causing different pathophysiological outcomes have been described (de Souza *et al*, 2010).

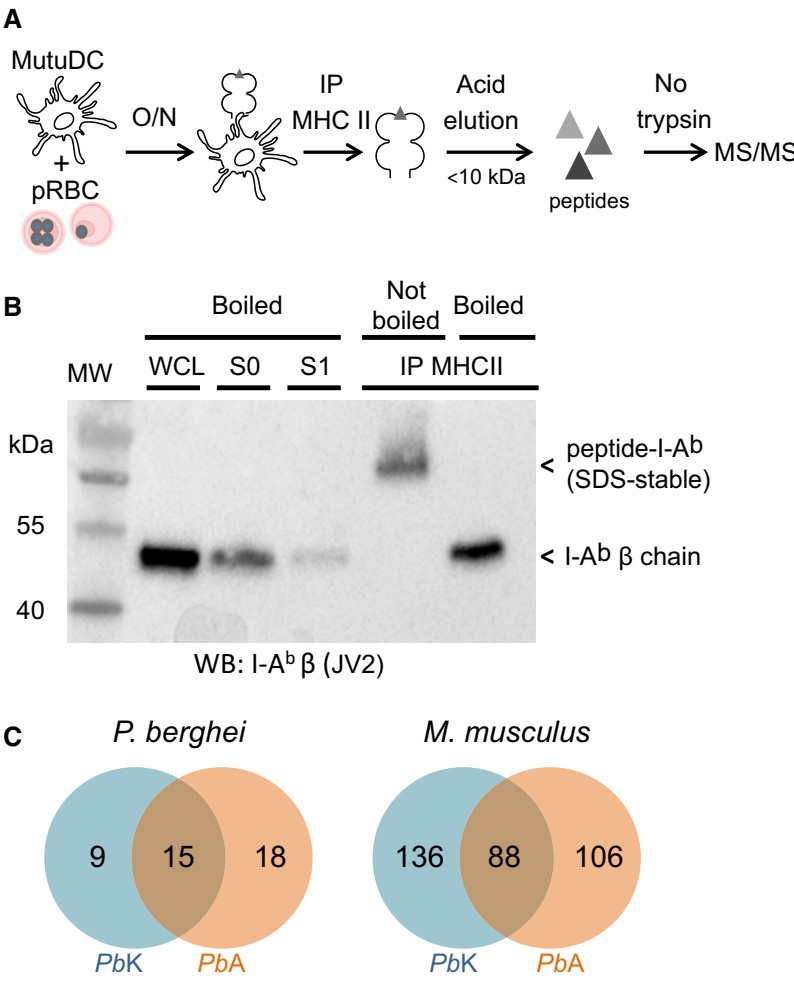

**Figure 1.  MS/MS profiling of *Pb*-derived MHC II immunopeptidome during blood-stage malaria.**

A   Experimental protocol used to immunoprecipitate MHC II molecules from *Pb*K or *Pb*A pRBC-loaded MutuDC and to elute MHC II-bound peptides before proteomic analysis.

B   Western blot revealed with anti-I-A[b]β JV2 antibody. WCL: whole-cell lysate of MutuDC + pRBC; S0: supernatant following incubation with protein-G beads; S1: 1st wash of protein-G beads; IP MHC II: eluates from Y-3P-immunoprecipitated MHC II, boiled or not. Representative of two independent experiments.

C   Venn diagram of unique peptides from *P. berghei* (left) and *Mus musculus* (right) identified in the experiments performed with *Pb*K pRBC (blue) and *Pb*A pRBC (orange).

*Pb* K173 (*Pb*K) being the most "distant" from *Pb*A (although SNPs were found only in 469 of > 15,000 genes; Otto *et al*, 2014), we decided to perform this analysis both with *Pb*A and *Pb*K pRBC. This led to the identification of a total of 372 unique I-A[b]-eluted peptides, comprising 330 mouse sequences and 42 *Pb* sequences (Fig 1C, Dataset EV1), derived from 13 putative *Pb* antigenic proteins (Dataset EV2). In line with the modest level of polymorphisms between *Pb*A and *Pb*K, none of those source antigens was polymorphic. Remarkably, for seven antigens, we recovered multiple peptides containing a core of 11–16 amino acids (aa) and various N- or C-terminal extensions (Dataset EV2). This feature is a typical property of MHC II ligands (Bozzacco *et al*, 2011; Sofron *et al*, 2016). For each set of nested peptides, we further analyzed the longest spanning sequence comprising all recovered sequences, for example, NALYNYSIPR PNVTSNL for ETRAMP10.2 (PBANKA_0517000) or LHASPYVA PAAAIIEMAE for LDH (PBANKA_1340100), resulting in a

consolidated list of 14 peptides (Table 1). We evaluated the I-A[b] binding affinity of these 14 peptides by quantifying their ability to inhibit binding of a high-affinity radiolabeled peptide (Sidney *et al*, 2013). Ten peptides were found to be good I-A[b] binders (IC$_{50}$ < 1,000 nM), three moderate binders (IC$_{50}$ < 5,000 nM), and one low binder (IC$_{50}$ > 5,000 nM; Table 1).

In summary, our analysis of the I-A[b] immunopeptidome in DC identifies a panel of 14 MHC II ligands derived from 13 non-polymorphic antigenic proteins expressed by *Pb*A and *Pb*K.

### *In vivo* relevance of the peptide panel during blood-stage malaria

In order to validate this panel *in vivo*, we measured the frequency of CD4 T cells responding to each peptide during *Pb*A pRBC infection (Fig 2A). Splenic T cells were restimulated with DC loaded with an irrelevant peptide (OVA) or uninfected RBC as negative controls,

**Table 1.  List of synthetic peptides tested in this study and their binding affinity to I-A$^b$.**

| Gene ID | Gene description | Synthetic spanning peptide | Location in the protein | Short peptide name | I-A$^b$ binding IC50 (nM) |
|---|---|---|---|---|---|
| PBANKA_0915000 | Apical membrane antigen 1 precursor | INDRNFIATTALSSTEE | 365–381 | AMA1 | **818** |
| PBANKA_0621900 | Mitochondrial import inner membrane translocase subunit TIM14 | GGSTYIAAKVNEAKD | 96–110 | TIM14 | **302** |
| PBANKA_1214300 | Enolase | TTLGIFRAAVPSGASTG | 28–44 | ENO | **115** |
| PBANKA_1326400 | Glyceraldehyde 3-phosphate dehydrogenase | GINHEKYNSSQTIVSNAS | 135–152 | GAPDH.1 | 2,420 |
| PBANKA_1145900 | Membrane-associated histidine-rich protein 1b | TEVPSLVPPTTNTSHAAPAH | 249–268 | MAHRP | **872** |
| PBANKA_1133300 | Elongation factor 1-alpha | SGKVVEENPKAIKSGDS | 369–385 | EF1a | 2,701 |
| PBANKA_1410300 | m1-family aminopeptidase | LSEVVIHPETNYALTG | 266–281 | M1 | 15,941 |
| PBANKA_0831000 | Merozoite surface protein-1 | APSEQTTTPEAATAASN | 984–1,000 | MSP1 | 1,346 |
| PBANKA_1340100 | L-lactate dehydrogenase | LHASPYVAPAAAIIEMAE | 231–248 | LDH | **110** |
| PBANKA_0517000 | Early transcribed membrane protein (ETRAMP10.2) | NALYNYSIPRPNVTSNL | 272–288 | ETRAMP | **87** |
| PBANKA_1450300 | ATP Synthase subunit beta mitochondrial | DNEYDFSGKAALVYGQ | 261–277 | ATPSYN | **104** |
| PBANKA_0505600 | Endomembrane prot 70 | KILYNSAKPNSDLH | 172–185 | END70 | **478** |
| PBANKA_0702800 | Protein disulfide isomerase | LIPEYNEAAIMLSEKK | 65–80 | PDI | **166** |
| PBANKA_1326400 | Glyceraldehyde-3-phosphate dehydrogenase | LMTTVHASTANQLVV | 175–189 | GAPDH.2 | **980** |

In the last column, IC$_{50}$ values lower than 1,000 nM (corresponding to good I-A$^b$ binders) are shown in bold.

and with *Pb*A pRBC as positive control. The latter condition allowed estimating the abundance of total parasite-specific CD4 T cells in the spleen, regardless of their peptide specificity. We chose a 10:1 ratio of pRBC to DC (Fig EV1A), and we focused on day 6 post-infection (corresponding to ECM onset) since the parasite-specific CD4 T-cell response was maximal at this time point and the mice succumbed beyond this day (Fig EV1B and C). In accordance with earlier studies (Villegas-Mendez *et al*, 2012), we observed a substantial "spontaneous" release of IFNγ by activated CD4 T cells (CD4$^+$ CD11a$^+$ CD49d$^+$) regardless of *ex vivo* re-exposure to antigen (Fig 2B). In order to improve the specificity of detection of genuine parasite-specific CD4 T cells, we focused on double IFNγ/TNF-producing cells. Thirteen of 14 peptides elicited a higher IFNγ/TNF production than the OVA peptide, with eight showing statistical significance. The three most dominant peptides originated from ETRAMP10.2 (NALYNYSIPRPNVTSNL, NL17), GAPDH (GINHEKYNSSQTIVSNAS, GS18), and EF1α (SGKVVEENPKAIKSGDS, SS17) proteins (Fig 2C). In total, CD4 T cells specific for those three peptides comprised more than one-third of the entire *Pb*A-specific response (Fig 2D). In line with the inability of *Pb*K pRBC to induce ECM at a 10$^6$ pRBC inoculum (data not shown), the proportion of IFNγ/TNF-producing CD4 T cells in *Pb*K-infected mice was lower. Yet the three most dominant responses were conserved overall (Fig EV2A). This peptide panel was also relevant in two other malaria models, in which protection strongly relies on humoral responses. In the self-resolutive *Pcc* model, six of 10 peptides tested elicited IFNγ/TNF-producing CD4 responses at day 6 post-infection (Fig EV2B). Note that the identified ETRAMP and MSP1 peptides are not expressed by *Pcc* due to sequence polymorphisms, hence the absence of reactivity. Another model of interest is the genetically attenuated parasite (GAP) *P. berghei* NK65 which lacks the histamine-releasing factor (*Pb* NK65 ΔHRF). This vaccine strain, in which the sequences of all identified peptides are conserved (Otto *et al*, 2014), causes self-resolving blood-stage infection and results in long-lasting

cross-stage and cross-species immunity (Demarta-Gatsi *et al*, 2016). Three weeks post-challenge with *Pb* NK65 ΔHRF, we could detect CD4 T cells reactive against 11 of the identified peptides (Fig EV2C).

While injection of pRBC is a widely used model to induce blood-stage malaria, natural infection normally starts with the inoculation of mosquito-derived sporozoites (spz) followed by pre-erythrocytic stages. To investigate the relevance of the peptide panel in blood-stage responses developing after pre-erythrocytic stages, we infected mice with *Pb*A spz. Since liver stages last ~48 h in rodents, we analyzed the CD4 responses at day 8 pi (Fig 3A), which is also the time of ECM onset in this model (Fig EV1C). We first noticed that the amplitude of the total parasite-specific CD4 response (as indicated by the pRBC-loaded DC condition, Fig 3B) was ~fourfold lower than at day 6 post-infection with pRBC (see Fig 2C). Furthermore, the immunodominance hierarchy slightly differed and this time, the immunodominant response was targeted to the GS18 peptide from GAPDH (Fig 3B), which was the 2$^{nd}$ dominant peptide after pRBC challenge. Immunodominance can be driven by different parameters, but antigen abundance usually plays a major role. Given that RNA-seq analysis at blood stage showed a similar and relatively high mRNA expression for ETRAMP10.2, GAPDH, and EF1α (>4,000 RPKM summed on all blood stages; Otto *et al*, 2014), we assumed that the different immunodominance profile may be caused by an earlier onset of GAPDH and EF1α expression during pre-erythrocytic stages. We also hypothesized that the ETRAMP-specific response might inflate later during blood stage developing after spz infection. To test this possibility, we treated spz-infected mice at days 6 and 7 with chloroquine (CQ) in order to prevent ECM mortality but without totally blunting blood-stage development (Fig 3A and C). We could then analyze the CD4 responses at day 14 pi, when parasitemia had reached a similar level as day 8 without CQ (Fig 3C). In these conditions, the ETRAMP peptide was immunodominant (Fig 3D).

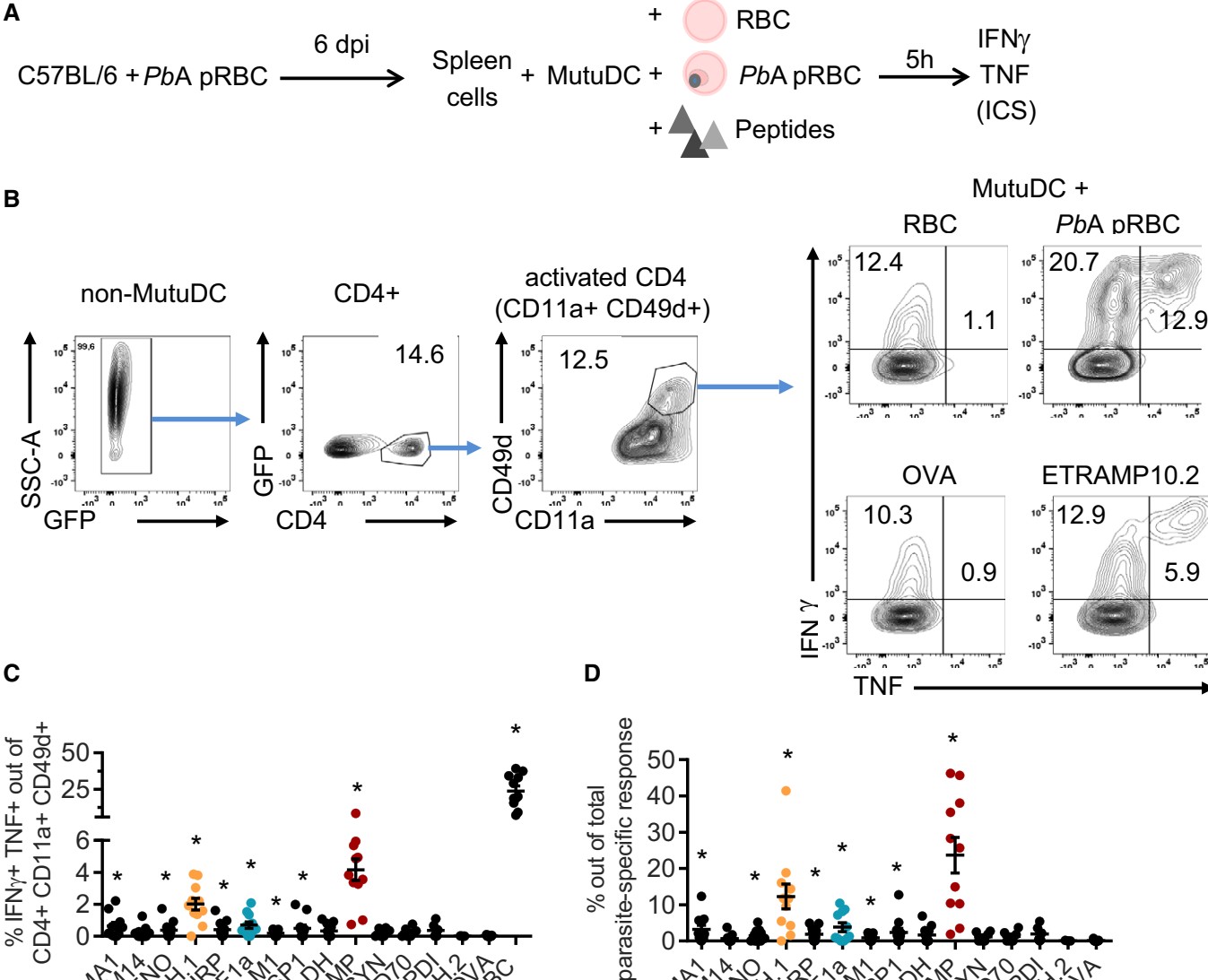

**Figure 2. Immunodominance of *Pb*A-derived MHC II peptides during blood-stage malaria induced after pRBC infection.**

A  Experimental scheme to analyze splenic CD4 T-cell responses at day 6 following inoculation of *Pb*A pRBC.

B  Gating strategy to calculate the proportion of CD11a⁺ CD49d⁺-activated CD4 T cells that produce IFNγ and TNF in response to MutuDC loaded with RBC, *Pb*A pRBC, OVA peptide (IR17), or ETRAMP10.2 peptide (NL17).

C  Percentage of IFNγ/TNF-double-producing cells among activated CD4 T cells for each peptide. Basal level with MutuDC alone was subtracted.

D  Hierarchy of immunodominance, depicted as the percentage of each peptide-specific response with respect to the total parasite-specific response obtained with *Pb*A pRBC-loaded MutuDC.

Data information: In (C, D), data show the mean ± SEM. Asterisks show statistical significance assessed by paired nonparametric Wilcoxon tests in comparison with OVA peptide. Panel (C): AMA1, *P = 0.0098; TIM14, P = 0.078; ENO, *P = 0.016; GAPDH.1, *P = 0.002; MAHRP, *P = 0.0039; EF1α, *P = 0.0039; M1, *P = 0.016; MSP1, *P = 0.031; LDH, P = 0.062; ETRAMP, *P = 0.001; ATPSYN, P = 0.25; END70, P = 0.25; PDI, P = 0.16; GAPDH.2, P = 0.99; pRBC, *P = 0.001. Panel (D): AMA1, *P = 0.0098; TIM14, P = 0.16; ENO, *P = 0.016; GAPDH.1, *P = 0.002; MAHRP, *P = 0.0039; EF1a, *P = 0.0039; M1, *P = 0.016; MSP1, *P = 0.031; LDH, P = 0.062; ETRAMP, *P = 0.001; ATPSYN, P = 0.25; END70, P = 0.25; PDI, P = 0.16; GAPDH.2, P = 0.99. N = 6 mice for ATPSYN, END70, PDI, GAPDH.2, N = 11 mice for all other peptides, pooled from three replicates.

In summary, our results indicate (i) that CD4 immunodominance hierarchy during blood stage is dictated in part by the route of challenge and by the duration of blood-stage infection and (ii) that ETRAMP ultimately establishes as a prominent blood-stage peptide after both spz and pRBC infections.

**cDC1 are superior to cDC2 for I-Aᵇ presentation of malaria antigens**

Next, we wanted to address which APC subset(s) control the generation of parasite-specific CD4 T cells by presenting malaria antigens

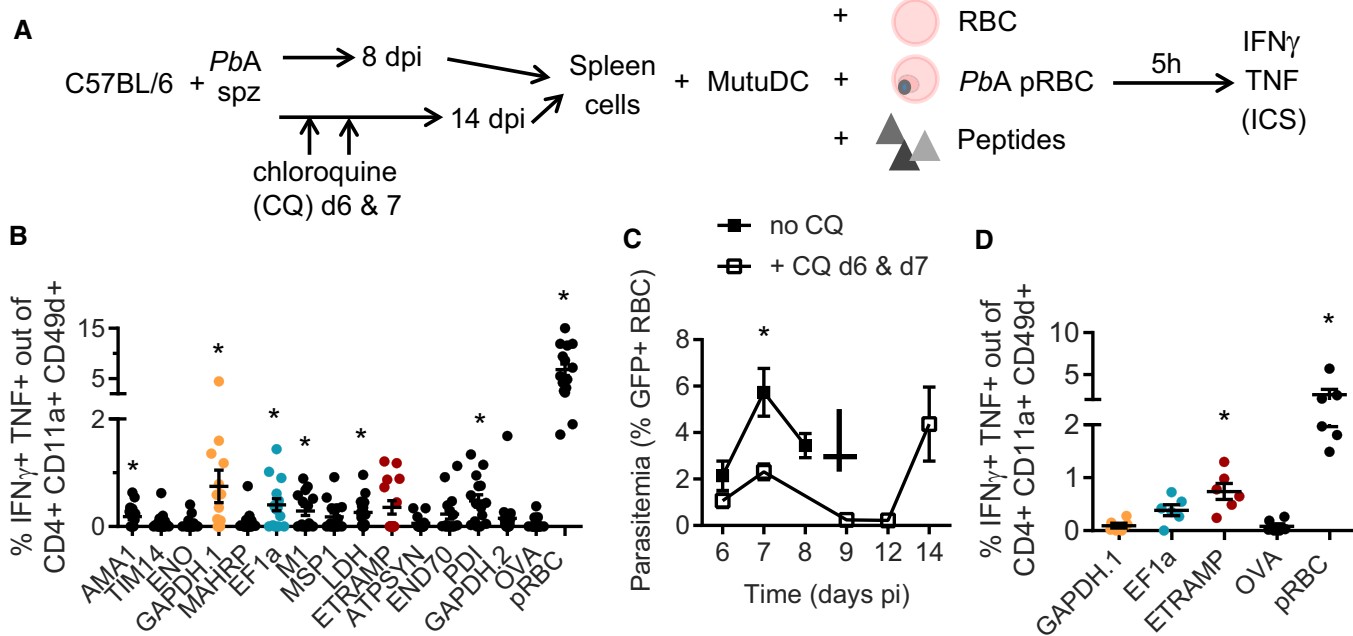

**Figure 3.  Immunodominance of *Pb*A-derived MHC II peptides during blood-stage malaria induced after spz infection.**

A   Experimental protocol used to analyze the *ex vivo* T-cell responses in the spleen at day 8 and 14 following spz inoculation, with or without chloroquine (CQ) treatment as indicated.

B   IFNγ/TNF-double-producing cells among activated CD4 T cells, after subtracting the basal level with MutuDC alone (mean ± SEM). Asterisks show statistical significance assessed by paired nonparametric Wilcoxon tests in comparison with OVA peptide. AMA1, *P = 0.027; TIM14, P = 0.84; ENO, P = 0.84; GAPDH.1, *P = 0.002; MAHRP, P = 0.71; EF1α, *P = 0.001; M1, *P = 0.032; MSP1, P = 0.23; *LDH, P = 0.016; ETRAMP, P = 0.064; ATPSYN, P = 0.77; END70, P = 0.097; PDI, *P = 0.0052; GAPDH.2, P = 0.19; pRBC, *P = 0.001. N = 15 mice pooled from three replicates.

C   Blood parasitemia (mean ± SEM) monitored by flow cytometry after inoculation of 5 × 10⁴ *Pb*A.GFP spz in B6 mice, treated (open squares) or not (black squares) with CQ at day 6 and 7 pi, in order to avoid ECM without blunting infection. Day 6, P = 0.29; Day 7, *P = 0.0013 by multiple unpaired *t*-tests.

D   IFNγ/TNF-double-producing cells among activated CD4 T cells, at day 14 pi following CQ treatment (mean ± SEM). Basal level with MutuDC alone was subtracted. GAPDH.1, P = 0.99; EF1α, P = 0.062; ETRAMP, *P = 0.031; pRBC, *P = 0.031 by paired nonparametric Wilcoxon tests in comparison with OVA peptide.

on MHC II. We set out to generate reporter CD4 T-cell hybridomas recognizing the dominant ETRAMP10.2 peptide, but as mentioned, MHC II ligands can be diverse in size (12–25 aa) and contain various N- and C-terminal extensions. To choose the optimal ETRAMP-derived peptide for raising the hybridomas, we assessed the ability of CD4 T cells from *Pb*A-infected mice to recognize N- and C-terminally extended versions of the NL17 peptide (Fig EV3A). As NL17 was among the most stimulatory peptides (Fig EV3B), we proceeded to generate NL17-specific T-cell hybridomas, that we named B6-derived CD4 T-cell hybrids reactive to ETRAMP and producing LacZ (BEZ). We confirmed that BEZ hybridomas reacted to *Pb*A pRBC- but not to uninfected RBC-loaded DC (Fig EV3C), and we observed that BEZ detected down to 2 nM of NL17 synthetic peptide (Fig EV3D).

To define the APC subset(s) presenting the ETRAMP10.2 peptide, we FACS-sorted three splenic DC subsets (cDC1, cDC2, pDC) and a 4th mixed population containing macrophages and monocyte-derived DC (moDC) from day 6-infected mice (Fig 4A) and used them in a BEZ antigen presentation assay. cDC1 were the most potent APC and were as stimulatory as *Pb*A pRBC-loaded MutuDC (Fig 4B). cDC2 were able to present the ETRAMP peptide albeit at much lower levels. We could not detect any presentation by pDC or by the macrophage/moDC mixed population (Fig 4B).

The lower performance of cDC2 cannot be explained by impaired I-A^b expression since I-A^b was slightly more abundant on cDC2 in day 6-infected mice (Fig 4C). Furthermore, when cDC1 and cDC2 were exogenously loaded with AS15, an I-A^b peptide from *T. gondii* (Grover *et al*, 2012), there was no difference in hybridoma stimulation at high peptide concentration (Fig 4D). The difference between cDC1 and cDC2 in Fig 4B is likely due to other factors, potentially including a different phagocytosis capacity and/or a different processing efficiency. Following incubation with heat-killed *T. gondii* (Fig 4E) and OVA-expressing *E. coli* bacteria (Fig 4F), cDC1 were indeed more potent for processing and presenting these exogenous antigens on MHC II. To evaluate the phagocytosis potential of these two DC subsets, we infected mice with *Pb*A.GFP and measured the proportion of GFP⁺ cells among cDC1. This proportion was about twice as high as that among cDC2, suggesting that cDC1 possess a better pRBC uptake capacity *in vivo* (Fig 4G). To analyze whether the selective cDC1 superiority was specifically related to severe malaria, we isolated cDC1 and cDC2 from naïve mice and exposed them to *Pb*A pRBC (Fig 4H) and to OVA-expressing *E. coli* (Fig 4I). In both contexts, cDC1 performed better than cDC2, and again, this was not due to a reduced MHC II surface expression (Fig 4J).

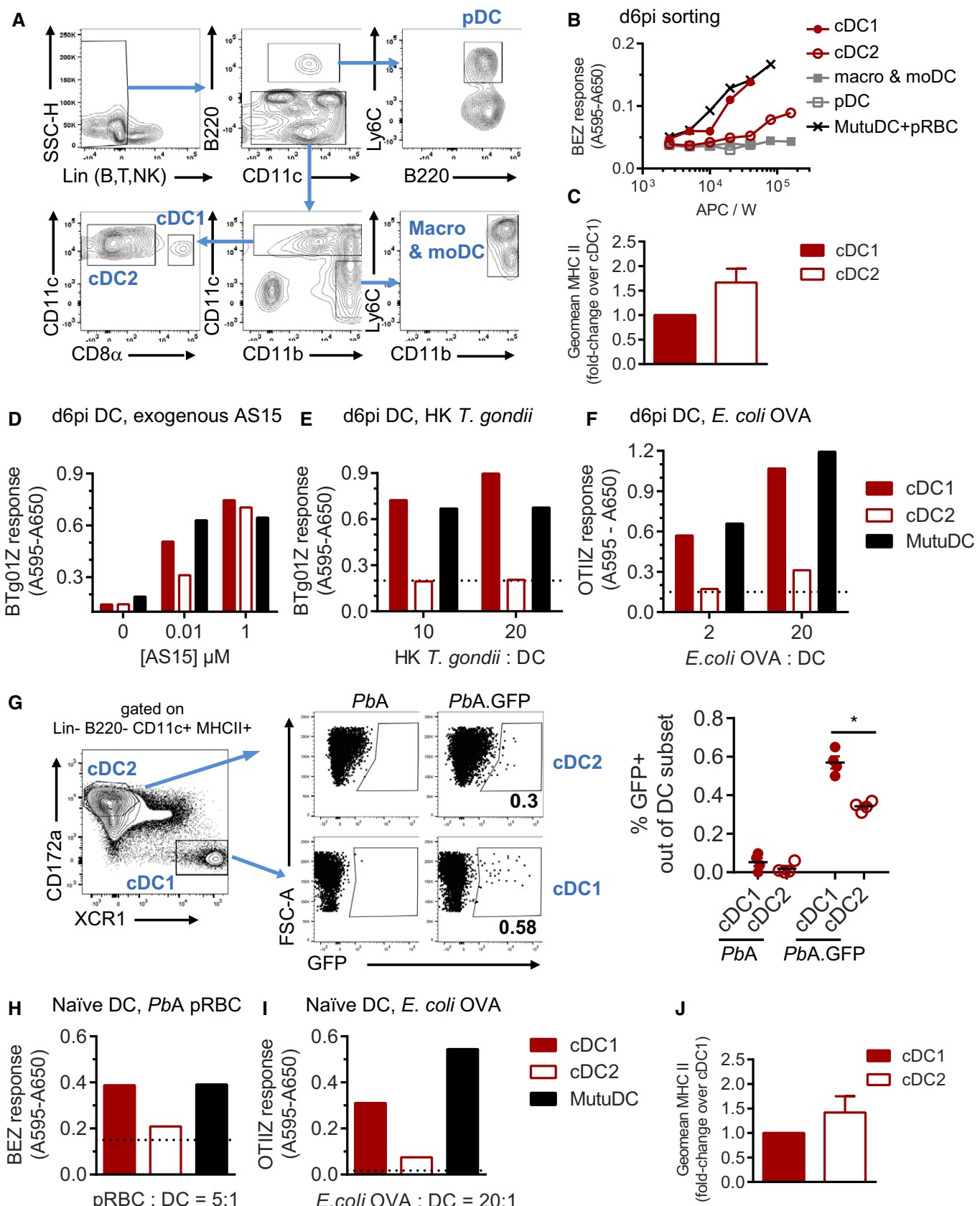

**Figure 4.**

**Figure 4.  cDC1 are superior to other spleen antigen-presenting cells for MHC II presentation of *Pb*A and non-*Pb*A antigens.**

A  Gating strategy for FACS-sorting pDC (Lin⁻ CD11c^lo B220⁺ Ly6C⁺), cDC1 (Lin⁻ CD11c⁺ CD8α⁺), cDC2 (Lin⁻ CD11c⁺ CD8α⁻), and a mix of macrophages and moDC (Lin⁻ CD11c⁻ CD11b⁺ Ly6C⁺) from spleens of mice infected for 6 days with *Pb*A pRBC.

B  I-A^b presentation of ETRAMP10.2 peptide by the indicated APC, measured with BEZ hybridomas. *Pb*A pRBC-loaded MutuDC were used as controls. Representative of two experiments with four infected mice pooled per replicate.

C  Fold-change surface expression of I-A^b in cDC2 over cDC1 in day 6-infected mice (mean ± SEM of five experiments). Difference is not significant based on unpaired *t*-test with Welch's correction.

D–F  I-A^b presentation of non-*Pb*A antigens by cDC isolated from day 6 *Pb*A-infected mice. (D) I-A^b-AS15 presentation after exogenous addition of AS15 peptide at the indicated concentrations, assessed with the BTg01Z hybridomas. (E) I-A^b-AS15 presentation by DC incubated with heat-killed *T. gondii* at the indicated ratio, assessed with the BTg01Z hybridomas. (F) I-A^b-IR17 presentation by DC incubated with OVA-expressing *E. coli* at the indicated ratio, assessed with the OTIIZ hybridomas. *Pb*A pRBC-loaded MutuDC were used as controls. Representative of two experiments with four infected mice pooled per replicate. Dotted line shows the basal response of the hybridomas in the absence of antigen.

G  *In vivo* pRBC uptake by cDC1 (XCR1⁺ CD172a⁻) and cDC2 (XCR1⁻ CD172a⁺) analyzed at 6 days post-infection with *Pb*A.GFP. Gates were positioned based on the auto-fluorescence of the cDC subset in day 6 *Pb*A-infected mice. As GFP⁺ events are rare, 10⁷ events were acquired for each mouse. Symbols show the mean percentage ± SEM (N = 4 mice per group) of GFP⁺ cells in each DC subset. *P = 0.0005 by unpaired *t*-test without assuming consistent SD.

H, I  I-A^b presentation by cDC isolated from naïve mice. (H) I-A^b-NL17 presentation by DC incubated with *Pb*A pRBC at a ratio of 5:1, assessed with the BEZ hybridomas. (I) I-A^b-IR17 presentation by DC incubated with OVA-expressing *E. coli* at a ratio of 20:1, assessed with the OTIIZ hybridomas. Representative of two independent experiments with 20 naïve mice pooled per experiment. Dotted line shows the basal response of the hybridomas in the absence of antigen.

J  Fold-change surface expression of I-A^b in cDC2 over cDC1 in naïve mice (mean ± SEM of four experiments). Difference is not significant based on unpaired *t*-test with Welch's correction.

Altogether, these data show that both in naïve mice and during severe *Pb*A malaria, cDC2 are selectively impeded in MHC II processing and presentation of pRBC-derived antigens as compared to cDC1, most likely leading to cDC1 playing prominent MHC II-related functions *in vivo*.

**cDC1 regulate CD4 T-cell functionality during severe malaria**

To decipher the *in vivo* implication of cDC1 in activating CD4 T cells, we took advantage of *Karma* mice that allow conditional depletion of cDC1 upon injection of DT (Alexandre *et al*, 2016). We first verified that DT treatment had no adverse effect on the development of parasitemia and on parasite-specific Th1 responses (data not shown). We started the DT treatment prior to infection and maintained the depletion throughout blood-stage development using repeated DT injections (Fig 5A). At day 6 pi, a clear reduction in CD8α⁺ DC was observed (Fig 5B and C) even though it appeared less pronounced than what is achieved at steady state (Alexandre *et al*, 2016).

Regardless of the cause for this partial depletion (e.g. influence of infection on cDC1 survival and/or differentiation), we confirmed a previous study reporting that cDC1 promote the development of *Pb*A-specific CD8 responses (Piva *et al*, 2012). The numbers of CD8 T cells responding to *Pb*A pRBC-loaded MutuDC, as well as to the F4 (Lau *et al*, 2011) and GAP50 (Howland *et al*, 2013) MHC I epitopes, were indeed substantially lower in DT-treated *Karma* mice (Fig EV4A). Most remarkably, cDC1 ablation resulted in dramatically impaired proportions (Fig 5D and E) and numbers (Fig 5F) of CD4 T cells co-producing IFNγ and TNF in response to *Pb*A pRBC-loaded or peptide-pulsed MutuDC. The same was true at day 4 post-infection (Fig EV4B) indicating that cDC1 are not only important for maintenance of IFNγ⁺ TNF⁺ Th1 cells but also for their priming. The effect of cDC1 depletion on the development of Th1 CD4 responses was not the mere consequence of altered parasite growth as blood parasitemia was comparable with and without DT treatment (Fig 5G). The reduced differentiation into IFNγ⁺ TNF⁺ CD4 T cells was also accompanied by further changes in the functionality of

**Figure 5.  cDC1 are required for the development of parasite-specific IFNγ⁺ TNF⁺ Th1 responses during *Pb*A blood-stage malaria.**

A  Experimental protocol to analyze T-cell responses in *Karma* mice at day 6 post-infection with *Pb*A pRBC. Mice were injected with DT prior to infection and repeatedly every 60 h thereafter.

B  Gating strategy to assess efficiency of cDC1 depletion.

C  Percentage (mean ± SEM) of CD8α⁺ out of Lin⁻ B220⁻ CD11c⁺ MHC II^hi cells. *P = 0.0043 by Mann–Whitney test. No DT group, N = 5 mice; + DT group, N = 6 mice. Representative of three independent experiments.

D  FACS plots showing the proportion of CD11a⁺ CD49d⁺ CD4 T cells that produce IFNγ and TNF in response to MutuDC loaded with RBC, *Pb*A pRBC, or ETRAMP10.2 peptide.

E  Proportion of CD11a⁺ CD49d⁺ CD4 T cells that co-produce IFNγ and TNF (mean ± SEM) in response to MutuDC loaded with the indicated antigen, isolated from *Karma* mice treated (open circles) or not (black circles) with DT. Asterisks show significant differences between no DT and + DT group by multiple unpaired *t*-tests without assuming consistent SD. RBC, P = 0.18; GAPDH.1, *P = 0.019; EF1a, *P = 0.039; ETRAMP, *P = 0.00054; OVA, P = 0.25; pRBC, *P = 0.015. No DT group, N = 5 mice; + DT group, N = 6 mice. Representative of three independent experiments.

F  Absolute numbers of IFNγ⁺ TNF⁺ CD11a⁺ CD49d⁺ CD4⁺ T cells responding to the indicated stimuli (mean ± SEM). GAPDH.1, *P = 0.0047; EF1a, *P = 0.00039; ETRAMP, *P = 0.00022; pRBC, *P = 0.00019 by multiple unpaired *t*-tests without assuming consistent SD. No DT group, N = 5 mice; + DT group, N = 6 mice. Representative of three independent experiments.

G  Blood parasitemia (mean ± SEM) at day 4 and 6 pi in *Karma* mice treated (open circles) or not (black circles) with DT. Day 4, *P = 0.022; Day 6, P = 0.77 by multiple unpaired *t*-tests without assuming consistent SD. No DT group, N = 5 mice; + DT group, N = 6 mice. Representative of three independent experiments.

H  Absolute numbers of CD4⁺ T cells producing IFNγ, TNF, or IL-10, either individually or in combination, following restimulation with MutuDC+*Pb*A pRBC (mean ± SEM). These cells are called cytokine⁺ in the next panels. *P = 0.0023 by unpaired *t*-test without assuming consistent SD. N = 3 mice per group.

I  Multifunctionality analysis (IFNγ, TNF, IL-10) of cytokine⁺ T cells responding to MutuDC loaded with *Pb*A pRBC. N = 3 mice per group.

J  Proportion of IL-10⁺ IFNγ⁻ TNF⁻ among cytokine⁺ T cells, in response to MutuDC loaded with the indicated antigen (mean ± SEM). GAPDH.1, P = 0.32; EF1a, P = 0.16; ETRAMP, **P = 8.5 × 10⁻⁵; OVA, P = 0.11; **pRBC, P = 0.0015 by multiple unpaired *t*-tests without assuming consistent SD. N = 3 mice per group.

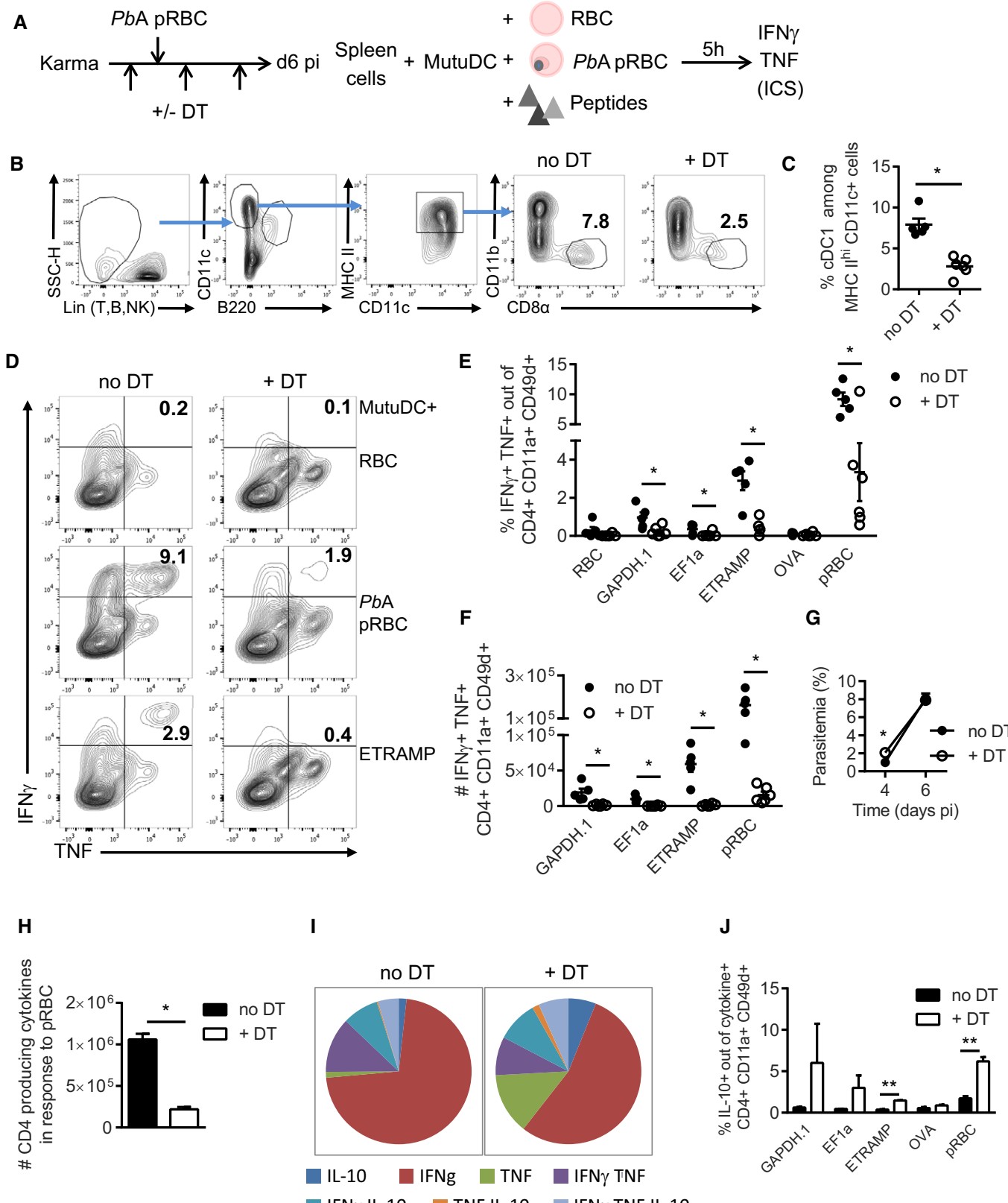

**Figure 5.**

parasite-specific CD4 T cells. Multifunctional analysis of IFNγ, TNF, IL-10 single, double, and triple producers performed on CD4 T cells responding to pRBC-loaded MutuDC (Fig 5H) showed an increased proportion of single IL-10$^+$ CD4 T cells in DT-treated mice (Fig 5I). Restimulation with the three most dominant peptides also led to higher proportions of IL-10$^+$ Tr1 cells in DT-treated mice (Fig 5J).

In conclusion, our data demonstrate that in the context of severe blood-stage malaria, cDC1 promote the differentiation of parasite-specific IFNγ$^+$ TNF$^+$ Th1 cells, to the expense of more regulatory IL-10$^+$ CD4 T cells.

## Discussion

In malaria-endemic areas, a nagging issue is the failure of naturally exposed individuals to develop sterile long-lasting protective immunity. This may be due to several factors including the stage specificity of parasite antigen expression, the antigenic variability among field parasites, and the profound immune dysregulation caused by pre-erythrocytic and erythrocytic stages (Renia & Goh, 2016; Scholzen & Sauerwein, 2016; Van Braeckel-Budimir et al, 2016). These factors could also contribute to explain why despite tremendous investments and years of research, progress on the vaccination front has been only modest. Clinical efficacy of the most advanced subunit vaccine candidate against *P. falciparum* (RTS,S) is limited and quickly wanes over time (Olotu et al, 2016). Hopes are emerging from whole attenuated sporozoite vaccination strategies (Sissoko et al, 2017), but the disappointing RTS,S results combined with the fact that vaccine research on *P. vivax* is only beginning (Tham et al, 2017) suggest the need to broaden our search for new subunit vaccine candidates and to dramatically improve the candidate down-selection process.

Mouse models of malaria are not faithful in every aspect to human malaria pathology, but they represent useful tools to establish proofs-of-concept, for example, related to immunomics approaches. Using *P. berghei* infection in B6 mice, our study shows a selective superiority of cDC1 in MHC II presentation of pRBC-associated antigens at ECM onset, that is, day 6 post-infection. DC functions are known to be impeded during blood-stage malaria (Cockburn & Zavala, 2016), and one can imagine that malaria may more strongly impact cDC2. However, our finding that cDC1 from naïve mice also present pRBC-derived antigens better than cDC2 suggests that this difference is not, or at least not solely, dictated by infection. Using a CD4 T-cell hybridoma that recognizes a currently undefined *Plasmodium* epitope, a Biorxiv-posted study (Fernandez-Ruiz et al, 2017) reports that cDC1 are more stimulatory than CD4$^+$ DC and double-negative DC, both in naïve mice and at 3 days post-infection. Although the sensitivity of our hybridomas did not allow us to evaluate the ETRAMP10.2 MHC II presentation at such an early time point (data not shown), our data are consistent with this work. What mechanism(s) could account for such a difference in MHC II presentation? One possibility is that cDC1 are equipped with a better pRBC uptake capability, which is supported by our evaluation of pRBC uptake in day 6-infected mice. In line with this idea, the phagocytic potential of cDC2 is specifically suppressed by type I IFN signaling during *Pb*A infection (Haque et al, 2014). Another non-mutually exclusive hypothesis is a difference in the efficiency of

the MHC II processing pathway. Phagolysosome maturation and/or hydrolase activity in MHC II processing compartments could be selectively enhanced in cDC1 and/or dysfunctional in cDC2. Alternatively, antioxidant pathways involving heme oxygenase-1, which are known to affect antigen presentation by DC (Riquelme et al, 2015), may be differentially elicited in cDC1 vs cDC2. Dissecting how malaria antigens are processed differently for MHC II presentation in DC subsets, and how severe malaria regulates this process, represents exciting avenues of future research.

A second substantial finding is that during severe malaria, cDC1 are critical for priming and promoting parasite-specific Th1 CD4 T cells. This is consistent with the results of a study using a CD4 TCR-transgenic mouse specific for a *Plasmodium* epitope (Fernandez-Ruiz et al, 2017). More generally, these data are in line with the established propensity of cDC1 to secrete IL-12 during infection by intracellular pathogens (Mashayekhi et al, 2011; Alexandre et al, 2016) and the implication of cDC1 in Th1 polarization during other parasitic infections such as leishmaniasis (Ashok et al, 2014; Martinez-Lopez et al, 2015) and toxoplasmosis (Mashayekhi et al, 2011). Perhaps most importantly, we find that cDC1 depletion, possibly by "forcing" MHC II presentation by the remaining cDC2 subset, skews cytokine production of parasite-specific CD4 T cells toward a more regulatory profile, associated with a higher proportion of single IL-10-producing CD4 T cells. This supports the emerging and prominent concept that functional polarization of CD4 Th cells is in part dictated by the subset(s) of cDC activating the CD4 T cells (Dutertre et al, 2014; Vu Manh et al, 2015). For instance, while IRF4 deficiency (resulting in impaired migration of cDC2) abrogates Th17 responses upon fungal challenge, Batf3 deficiency (resulting in lack of cDC1) skews CD4 responses toward Tregs and Th2, ultimately perturbing *Leishmania* parasite control (Ashok et al, 2014). In a non-infectious context, a mouse model of type I diabetes, targeting of a pancreatic cell antigen to cDC2 favors tolerogenic properties by autoreactive CD4 T cells as compared to cDC1 targeting (Price et al, 2015). Given the major regulatory role of IL-10 produced by CD4 Th cells during *Pcc* infection (Freitas do Rosario & Langhorne, 2012), one could speculate that the sustained antigen presentation by cDC2 in this setting (Sponaas et al, 2006) is involved in preventing immunopathology. In turn, one could hypothesize that addition of cDC2 or restoration of cDC2 MHC II presentation during *Pb*A infection may mirror cDC1 depletion and enhance the differentiation of regulatory Tr1 cells, which could ultimately alleviate ECM immunopathology. Harnessing DC subsets for immunotherapy of rheumatoid diseases (Pozsgay et al, 2017) and autoimmune pathologies (Liu & Cao, 2015; Audiger et al, 2017) is already a major research area. Modulating cDC1 to curtail excessive Th1 responses in human severe malaria could represent a valuable translational approach.

So far at least eight MHC I epitopes were characterized in B6 mice infected with *Pb*A (Lau et al, 2011, 2014; Howland et al, 2013; Poh et al, 2014), but no CD4 T-cell epitope was known. By characterizing the MHC II immunopeptidome by MS/MS, our study identifies a panel of MHC II ligands able to restimulate CD4 T cells isolated from *Pb*A-infected mice. *Pb*A infection is suited to evaluate the role of DC on T-cell polarization and pathogenicity during cerebral malaria, but this acute model is not well adapted to study the effects of CD4-dependent antibody responses on parasitemia. As previously reported (Piva et al, 2012), we observed no major effect

of cDC1 depletion on parasite growth. *Pcc* and *Pb* NK65 ΔHRF are likely to be better models to evaluate the relevance of the identified peptides on parasitemia and antibody-mediated protection. While our study does not analyze these aspects, we found that some peptides were recognized by CD4 T cells isolated from *Pcc*-infected mice and mice immunized with the *Pb* NK65 ΔHRF GAP, thus setting the stage to assess (i) whether the conserved peptides will elicit protection after challenge with *Pcc* and (ii) whether the protective antibody responses generated by the *Pb* NK65 ΔHRF vaccine GAP target the same MHC II antigens. Among the unraveled antigen panel, two of them may be particularly relevant to further analyze: ETRAMP10.2 and GAPDH. ETRAMP10.2 belongs to a family of proteins present in mouse and human *Plasmodium* that is secreted and anchored at the parasitophorous vacuole membrane (Spielmann *et al*, 2003; Pasini *et al*, 2013). Interestingly, ETRAMP10.2 and other members (ETRAMP2, 4 and 10.1) are recognized by immunoglobulins from individuals living in a high-endemicity malaria region of Papua New Guinea (Spielmann *et al*, 2003) and ETRAMP11.2 is recognized by sera from > 80% of *P. vivax*-exposed individuals (Chen *et al*, 2015), showing that ETRAMPs are relevant B-cell antigens in humans. GAPDH is primarily involved in glycolysis, but this enzyme also functions in processes ranging from DNA repair to membrane trafficking, iron transport, and cell signaling (Perez-Casal & Potter, 2016). Akin to other housekeeping proteins, it had long been assumed that GAPDH was present exclusively in the cytoplasm, but accumulating evidence suggests that it is also present at the surface of pathogens such as *Schistosoma mansoni* helminths (Argiro *et al*, 2000) and *Streptococcus* bacteria, where it could modulate virulence (Jin *et al*, 2011). Interestingly, resistance to *S. mansoni* reinfection was found to correlate with serum reactivity toward a GAPDH epitope (Argiro *et al*, 2000). In this context, GAPDH has been proposed as a vaccine candidate in several infectious settings and animal models (Perez-Casal & Potter, 2016). During malaria, GAPDH is expressed during asexual blood stage at the apical end of merozoites where it may be involved in membrane trafficking and vesicular transport (Daubenberger *et al*, 2003). Analysis of the surface-exposed spz proteome (Swearingen *et al*, 2016) and immunofluorescence assays (Cha *et al*, 2016) revealed that GAPDH is expressed on the surface of *P. falciparum* and of *P. berghei* spz. Notably, surface GAPDH interacts with CD68 expressed on Kupffer cells and is involved in liver invasion (Cha *et al*, 2016). Furthermore, immunization of Swiss-Webster mice with KLH-coupled GAPDH conferred protection after spz challenge (Cha *et al*, 2016). Knowing that the GAPDH GS18 epitope is conserved in *Pb*A, *Pb*K173, *Pb*NK65, *Py*17XNL, and *Pcc*, it seems relevant to further investigate the value of GAPDH as a multistage invasion-blocking vaccine candidate in pre-clinical mouse models. Our data also suggest analyzing whether natural antibody responses targeting *Plasmodium* GAPDH in humans are associated with a certain degree of immunity and/or clinical protection.

At last, our work provides a proof-of-concept supporting the relevance of profiling the HLA class II immunopeptidome for blood-stage *P. falciparum*. The biological and mass spectrometry techniques to analyze immunopeptidomes are rapidly evolving (Caron *et al*, 2015) and methods to characterize HLA DR-bound immunopeptidomes with relatively low numbers of cells have been optimized (Heyder *et al*, 2016), supporting the feasibility of this approach in human malaria. We expect this to lead to the identification of naturally processed epitopes recognized by CD4 T cells, including $T_{FH}$, in infected humans. CD4 T cell help to CD8 T cells is needed to generate robust memory CD8 T cells (Janssen *et al*, 2005), and the ability to generate fully functional $T_{FH}$ is critical for maintenance of effective long-term humoral immunity against malaria (Hansen *et al*, 2017). Thus, applying this approach in human malaria may both uncover new vaccine targets and help us down-select vaccine candidates that are more likely to drive robust antibody responses through $T_{FH}$ restimulation and long-lasting CD8 T-cell memory differentiation through DC-integrated CD4 help.

## Materials and Methods

### Ethics statement

Animal care and use protocols were carried out under the control of the National Veterinary Services and in accordance with the European regulations (EEC Council Directive, 2010/63/EU, September 2010). The protocol (CE no. 2015-03) was approved by the local Ethical Committee for Animal Experimentation (registered by the "Comité National de Réflexion Ethique sur l'Expérimentation Animale" under no. CEEA122).

### Mice, parasites, and experimental infections

C57BL/6J (B6) were purchased from Janvier (France), and *Karma* mice (*a530099j19rik-tm1Ciphe*; Alexandre *et al*, 2016) were imported from CIPHE, Marseille, France. All mice were males between 7 and 16 weeks old. They were housed under specific pathogen-free conditions at UMS006-CREFRE, Toulouse. *Plasmodium berghei* ANKA (*Pb*A) and *P. berghei* Kyberg 173 (*Pb*K) parasites were propagated in B6. To prepare pRBC stocks, mice were bled in heparin and the proportion of pRBC was evaluated by blood smear. pRBC concentration was adjusted to $10^7$ pRBC/ml in Grau solution (Alsever's solution with 10% glycerin), and 1 ml aliquots were stored at −80°C. All pRBC infections were done by intravenous inoculation of $10^6$ pRBC, unless otherwise indicated. Sporozoites were obtained from salivary glands of *Anopheles stephensi* mosquitoes infected with *Pb*A expressing GFP under the HSP70 promoter, as described (Manzoni *et al*, 2014). In brief, *A. stephensi* mosquitoes were reared at 26°C and 80% humidity, and fed on 10% sucrose solution. Adult mosquitoes from 3 to 7 days were fed on anesthetized *Pb*A-infected mice and further kept at 21°C. Salivary glands of *Pb*A-infected mosquitoes were removed by hand dissection 21–28 days post-feeding and crushed in Leibovitz's L-15 medium to release sporozoites. Parasites were counted and kept at 4°C until use. Mice were infected i.v. with $5 \times 10^4$ spz.

Parasitemia was measured by blood smear or flow cytometry with similar results. For flow cytometry, 3 μl of blood tail was collected using a microvette. With *Pb*A.GFP, the percentage of pRBC was determined by gating on GFP$^+$ out of total RBC on non-fixed blood. With non-fluorescent parasites, the blood was labeled with Ter119-FITC (1/30, Miltenyi Biotec), CD71-PE (C2, 1/300, BD Pharmingen) and CD41-PE-Cy7 (MWReg30, 1/100, Biolegend), fixed and permeabilized in 4% PFA and 0.6% saponin followed by DAPI staining. The percentage of pRBC was determined by gating on DAPI$^+$ out of Ter119$^+$ CD41$^-$ cells.

## MutuDC loading, I-A$^b$ immunoprecipitation, and peptide elution

pRBC from freshly harvested blood were enriched using Percoll in order to obtain > 70% parasitemia. $10^8$ MutuDC were incubated overnight with $10^9$ *Pb*A or *Pb*K pRBC at 37°C. Loaded DC were lysed using CHAPS buffer during 45 min at 4°C, as previously described (Bozzacco & Yu, 2013). The lysate was cleared by centrifugation at 14,000 *g* for 15 min, and MHC II molecules were immunoprecipitated from the cleared lysate using 50 µg of Y-3P antibody (BioX-Cell) bound to protein-G beads overnight at 4°C. Beads were washed 3 times, and peptides were eluted with 10% acetic acid at 70°C for 10 min.

## Western blot

Immunoprecipitation eluates were reduced in β-mercaptoethanol and boiled or not in Laemmli buffer. Lysates corresponding to $7.5 \times 10^5$ DC equivalents were separated by electrophoresis on 10% polyacrylamide gels and transferred to nitrocellulose membranes. Immunologic detection was achieved using primary rabbit anti-mouse I-A$^b$ (JV2, 1/1,000) followed by horseradish peroxidase-conjugated mouse anti-rabbit antibodies (Promega). Detection of peroxidase activity was achieved using a ChemiDoc system (Bio-Rad).

## Nano-LC-MS/MS analysis

Prior to analysis, peptides were separated from protein using a Vivaspin concentrator with 10-kDa MW cutoff. Peptides were subjected to solid-phase extraction using a C18 Sep-Pak cartridge and finally concentrated using a rotating evaporator. Nano-LC-MS/MS analysis was performed on an UltiMate 3000 RSLCnano System (Thermo Fisher Scientific) coupled to a Q-Exactive mass spectrometer (Thermo Fisher Scientific). Peptides were automatically fractionated onto a C18 reverse-phase column (75 µm × 150 mm, 2 µm particle, PepMap100 RSLC column, Thermo Fisher Scientific) at a temperature of 35°C. Trapping was performed during 4 min at 5 µl/min, with solvent A (98% $H_2O$, 2% ACN and 0.1% FA). Elution was performed using two solvents A (0.1% FA in water) and B (0.1% FA in ACN) at a flow rate of 300 nl/min. Gradient separation was 36 min from 2% B to 40% B, 2 min to 90% B, and maintained for 5 min. The column was equilibrated for 13 min with 5% buffer B prior to the next sample analysis. The electrospray voltage was 1.9 kV, and the capillary temperature was 275°C. Full MS scans were acquired over m/z 400–2,000 range with resolution 70,000 (m/z 200). The target value was $10^6$. Ten most intense peaks with charge state between 2 and 7 were fragmented in the HCD collision cell with normalized collision energy of 35%, and tandem mass spectrum was acquired with resolution 35,000 at m/z 200. The target value was $2 \times 10^5$. The ion selection threshold was $6.7 \times 10^4$ counts, and the maximum allowed ion accumulation times were 250 ms for full MS scans and 150 ms for tandem mass spectrum. Dynamic exclusion was set to 30 s.

## Proteomic data analysis

Raw data collected during nano-LC-MS/MS analyses were processed and converted into *.mgf peak list format with Proteome Discoverer 1.4 (Thermo Fisher Scientific). MS/MS data were interpreted using search engine Mascot (version 2.4.1, Matrix Science, London, UK) installed on a local server. Searches were performed with a tolerance on mass measurement of 0.02 Da for precursor and 10 ppm for fragment ions, against a composite target decoy database (189,576 total entries) built with *Plasmodium berghei* UniProt database (TaxID = 5,823, December 3, 2015, 13,853 entries), *Mus musculus* UniProt database (TaxID = 10,090, December 3, 2015, 78,986 entries) fused with the sequences of recombinant trypsin and a list of classical contaminants (117 entries). Methionine oxidation and protein N-terminal acetylation were searched as variable modifications, and no enzyme was indicated. For each sample, peptides were filtered out according to the cutoff set for proteins hits with 1 or more peptides taller than nine residues and ion score > 19, allowing a false-positive identification rate of 2.5% for protein and 0.4% for peptides.

## MHC purification and binding assays

Purification of I-A$^b$ MHC II molecules by affinity chromatography and the assay based on the inhibition of binding of a high-affinity radiolabeled peptide to quantitatively measure peptide binding were described elsewhere (Sidney *et al*, 2013). Briefly, the mouse B-cell lymphoma LB27.4 was used as a source of MHC molecules. A high-affinity radiolabeled peptide (0.1–1 nM; peptide ROIV, sequence YAHAAHAAHAAHAAHAA) was co-incubated at room temperature with purified MHC in the presence of a cocktail of protease inhibitors and an inhibitor peptide. Following a 2-day incubation, MHC-bound radioactivity was determined by capturing MHC/peptide complexes on mAb (Y3JP)-coated Lumitrac 600 plates (Greiner Bio-one), and measuring bound cpm using the TopCount (Packard Instrument Co) microscintillation counter. The concentration of peptide yielding 50% inhibition of the binding of the radiolabeled peptide was calculated. Under the conditions utilized, where [label] < [MHC] and IC50 ≥ [MHC], the measured $IC_{50}$ values are reasonable approximations of the true $K_d$ values. Each competitor peptide was tested at six different concentrations covering a 100,000-fold range. The "cold" probe was used as a positive control in each experiment.

## Generation of reporter CD4 T-cell hybridomas

Mice were infected with *Pb*A pRBC and treated with intraperitoneal injections of 0.4 mg of chloroquine (CQ) at days 3, 4, 5, and 6 pi. Spleens were harvested at day 7 pi, and $6 \times 10^6$ splenocytes were seeded per well into a P24 in complete RPMI medium supplemented with 1 µM of ETRAMP NL17 peptide. After 2 days, 50 U/ml of recombinant human IL2 (rhIL2, BD Biosciences) was added. After 1 week, cells were separated by Ficoll and CD4 T cells were enriched using magnetic sorting (Miltenyi). CD4 T cells were tested for reactivity and put back in culture with 35 Gray-irradiated MutuDC, 1 µM of NL17, 50 U/ml rhIL2, and 5% of T-stim solution. Three days after, T-cell cultures were subjected to Ficoll separation and fused with the TCRαβ-negative lacZ-inducible BWZ fusion partner, as previously described (Feliu *et al*, 2013). Hybridomas were selected and subcloned at least 3 times. The reactivity was tested by mixing $10^5$ hybridomas with $5 \times 10^4$ MutuDC or BMDC, previously loaded with varying ratios of pRBC or RBC extracts overnight, or

incubated with serially diluted NL17 peptide. RBC extracts were obtained by three freeze/thaw cycles (liquid nitrogen/37°C).

TCR-triggered stimulation of the hybridomas was quantified using a chromogenic substrate: chlorophenol red-β-D-galactopyranoside (CPRG, Roche). Cleavage of CPRG by β-galactosidase releases a purple product, which absorbance was read at 595 nm with a reference at 650 nm.

### *Ex vivo* spleen T-cell restimulations

Spleens from infected mice were collected, mashed on a 70-μm cell strainer in RPMI supplemented with 10% of FBS and treated with RBC lysis buffer (ACK: 100 μM EDTA, 160 mM $NH_4Cl$, and 10 mM $NaHCO_3$). $10^6$ spleen cells were incubated for 5 h at 37°C in the presence of brefeldin A with $10^5$ DC with or without 5 μM of peptide, or with pRBC-loaded DC. Intracellular IFNγ (IFNγ-APC, XMG1.2 1/500 BD Pharmingen), TNF (TNF-AF700, MP6-XT22 1/300 BD Pharmingen), and IL-10 (IL10-BV650, JES5-16E3 1/300 BD Horizon) were detected with Cytofix/Cytoperm kit (BD Pharmingen) or Intracellular Fixation and Permeabilization Buffer Set (eBioscience). In addition to the newly identified *Pb*A peptides, we used the following control peptides: OVA/IR17 (ISQAVHAAHAEINEAGR) and *T. gondii* AS15 (AVEIHRPVPGTAPPS) restricted by I-A[b], F4 (EIYIFTNI; Lau *et al*, 2011), and GAP50 (SQLLNAKYL; Howland *et al*, 2013) restricted by K[b] and D[b,] respectively.

### Analysis and *ex vivo* isolation of spleen APC subsets

Spleens were perfused with collagenase D (1 mg/ml) and DNAse I (0.1 mg/ml), cut into small pieces and incubated at 37°C for 45 min. Cell preparations were filtered using 100-μM cell strainers, and APC populations were enriched using magnetic depletion of B, NK, and T cells (Miltenyi). For experiments that focused only on cDC, cDC were enriched using CD11c-positive selection (Miltenyi). Cell suspensions were labeled with the following antibodies: CD3-APC (145-2C11, 1/300 eBioscience), CD19-APC (ID3, 1/300 Biolegend), NK1.1-APC (PK136, 1/300 BD Pharmingen), B220-BV510 (RA3-6B2, 1/500, BD Horizon), CD11b PE-CF594 (M1/70, 1/3000 BD Horizon), CD11c PE-Cy7 (HL3, 1/400 BD Pharmingen), Ly6C-PerCP-Cy5.5 (AL-21, 1/1000 BD Pharmingen), CD64 BV421 (X54-5/7.1, 1/300 Biolegend), CD8α APC-Cy7 (53-6.7, 1/200 BD Pharmingen), or XCR1-PE (REA707, 1/100 Miltenyi). Cells were sorted on a BD AriaSorp (BD Biosciences) with purity routinely higher than 95%. Expression of MHC II was assessed in a separate mix since the anti-I-A[b] AF700 (M5/114, 1/500 BD Pharmingen) is a blocking antibody. For experiments with *Karma* mice in which cDC1 are Tomato[+], CD11b-PerCP-Cy5 (M1/70, 1/200 BD Pharmingen) was used. All flow cytometry samples were run on a Fortessa (BD Biosciences) and analyzed using FlowJo software.

### *Ex vivo* antigen presentation assays

FACS-sorted APC were serially diluted into 96-W round-bottom plates and mixed with $10^5$ T-cell hybrids/well. For non-*Pb*A antigens, $5 \times 10^4$ FACS-sorted DC per well were seeded in a 96-W round-bottom plate and treated as follows. To readout presentation of heat-killed *T. gondii*, RH tachyzoites were treated for 15 min at 56°C in PBS and incubated with DC for 3 h before adding BTg01Z

**The paper explained**

**Problem**

Malaria is a parasitic disease causing life-threatening complications like cerebral malaria. Natural protective immunity takes years to develop, and the efficacy of currently developed vaccines is limited. During malaria, the immune cross talk between dendritic cells (DC) and T cells prompts CD4 T cells to differentiate into functionally distinct subsets, which control the fate of memory CD8 T cell and antibody responses and ultimately determine the balance between parasite control and immunopathology. A major aim is to improve protective immunity without exacerbating immunopathology. To do this, one needs to elucidate how DC control the functionality of parasite-specific CD4 T cells.

**Results**

We work with a mouse malaria model induced by *Plasmodium berghei*. We use proteomics to identify parasite MHC class II antigens that elicit CD4 T cells in infected mice, as well as in long-term protected mice vaccinated with an attenuated parasite. We find that a peculiar DC subset, named cDC1, is superior for MHC II presentation of *Plasmodium*-derived antigens. We further show that cDC1 promote the differentiation of a type of parasite-specific CD4 T cells that play a pathogenic role in cerebral malaria (Th1) and that they inhibit the development of regulatory IL-10[+] CD4 T cells.

**Impact**

Our work sets the stage for MHC II immunopeptidome profiling in blood-stage human malaria. This should help uncover new vaccine targets and down-select vaccine candidates. Our study also supports the idea of harnessing DC functions to curtail excessive Th1 responses in human severe malaria.

hybrids (Grover *et al*, 2012). To readout presentation of OVA-expressing *E. coli*, an overnight pre-culture of DH5α transformed with pGEX.4T1.OVA was transferred to 7 ml of LB medium containing 100 μg/ml ampicillin and grown for 1 h. IPTG was added, and bacteria were grown for 4 more hours at 30°C. Bacteria were counted, washed, suspended in antibiotic-free serum-free RPMI, and incubated with DC at the indicated MOI for 40 min. After washing, gentamycin (100 μg/ml) was added for 3 h followed by incubation with $10^5$ OTIIZ/well (Sahara & Shastri, 2003). For exogenous peptide loading, the *T. gondii* I-A[b] peptide (AS15) was added to the DC simultaneously with the BTg01Z hybrids.

### Statistical analyses

The Prism software (GraphPad) was used for statistical analyses. To compare each peptide to the OVA peptide in *ex vivo* restimulation assays, nonparametric paired *t*-tests (Wilcoxon) were used. For other comparisons, unpaired *t*-tests without assuming consistent SD (Welch's correction) were used.

### Data deposition

The mass spectrometry proteomics data have been deposited to the ProteomeXchange Consortium via the PRIDE partner repository with the dataset identifier PXD007628.

**Expanded View** for this article is available online.

## Acknowledgements

F. L'Faqihi-Olive, V. Duplan-Eche, A.-L. Iscache, and P. Menut for technical assistance at the CPTP-INSERM U1043 flow cytometry facility, R. Balouzat and the zootechnicians at INSERM UMS006-CREFRE mouse facility, J.F. Franetich, M. Tefit, and T. Houpert for mosquito breeding, E. Bassot and F. de Giorgi for technical help, the Blanchard team for helpful discussions, H. Acha-Orbea for the MutuDC, M. Rescigno for the *E. coli*-OVA, A.-M. Lennon-Duménil and D. Lankar for the JV2 antibody. This work was supported by Human Frontier Science Program Organization (CDA00047/2011 to NB), PIA PARAFRAP Consortium (ANR-11-LABX0024 to NB and OS), PIA ANINFIMIP equipment (ANR-11-EQPX-0003 to NB), Région Occitanie PhD co-funding (MD), "Ministère de l'Education Nationale, de la Recherche et de la Technologie" (AH), Idex Toulouse "Attractivity Chair" Program (NB).

## Author contributions

Conceived and designed the experiments: MDr, MFW, J-MS, TDA, ST, JS, ASe, AB, OS, NB. Performed the experiments: MDr, MFW, J-MS, TDA, AH, ASa, JS. Provided *Karma* mice and helped design the related experiments: KC, MDa. Provided mice immunized with *Pb* NK65 ΔHRF: CD-G. Wrote the manuscript: NB with help of coauthors.

## Conflict of interest

The authors declare that they have no conflict of interest.

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
