## [Review Process File · EMBO Molecular Medicine]

Profiling MHC II immunopeptidome of blood stage malaria reveals that cDC1 control the functionality of parasite-specific CD4 T cells

Marion Draheim, Myriam F Wlodarczyk, Karine Crozat, Jean-Michel Saliou, Tchilabalo Dilezitoko Alayi, Stanislas Tomavo, Ali Hassan, Anna Salvioni, Claudia Demarta-Gatsi, John Sidney, Alessandro Sette, Marc Dalod, Antoine Berry, Olivier Silvie, Nicolas Blanchard

Corresponding author: Nicolas Blanchard, INSERM

Review timeline:

Submission date:	08 June 2017
Editorial Decision:	18 July 2017
Revision received:	04 August 2017
Editorial Decision:	16 August 2017
Revision received:	25 August 2017
Accepted:	29 August 2017

Transaction Report:

Editor: Céline Carret

1st Editorial Decision

18 July 2017

Thank you for the submission of your manuscript to EMBO Molecular Medicine and for your patience. We have now heard back from the two referees whom we asked to evaluate your manuscript.

As you will see from the comments below, the referees are enthusiastic about the study but do have suggestions and recommendations to further improve conclusiveness and clarity. In addition, we would like you to discuss more your findings with clinical and translational implications in mind, as this is particularly important for our scope.

We would welcome the submission of a revised version within three months for further consideration and would like to encourage you to address (experimentally when needed) all the criticisms raised as suggested to improve conclusiveness and clarity. Please note that EMBO Molecular Medicine strongly supports a single round of revision and that, as acceptance or rejection of the manuscript will depend on another round of review, your responses should be as complete as possible.

I look forward to receiving your revised manuscript.

***** Reviewer's comments *****

Referee #1 (Comments on Novelty/Model System):

The science in this report is excellent. The identification of peptide epitopes for MHC II-restricted antigens of *P. berghei* is novel and is the major finding of this report. The effect of depletion of DC1 on the response of CD4 T cells of defined specificity is also novel. The medical impact is distant as it is mainly allowing greater exploration of the mouse model in more detail, for eventual translation. Though, it also provides insight into the role of DC1 in priming of CD4 T cell immunity, which is likely to translate to humans. The *P. berghei* model is an adequate model for examining immunity to malaria, though all animal models have their nuances.

Referee #1 (Remarks):

The report by Draheim and colleagues undertakes a strategy to identify *P. berghei* peptides presented in the context of IAb during blood-stage malaria. It successfully identifies a number of peptide antigens that are able to stimulate CD4 T cells responding to this infection. The authors then generate a hybridoma against one of these peptides and examine which DC subsets are presenting antigen at day 6 of infection. This reveals that cDC1 are the primary presenting population. DC2 are shown to be somewhat defective in antigen presentation at this stage. Depletion of the DC1 population throughout infection reduces the overall CD4 T cell response to *P. berghei* and results in skewing of the remaining response IFN γ /TNF production (Th1) to IL-10 production (Tr1). This suggests DC1 are important in generation of a Th1 biased response in this setting. The experiments are well performed and that findings are valuable and interesting. The identification of MHC II-restricted epitopes will allow greater exploration of this response in future studies.

I have only one additional set of experiments that would enhance this study. It would be useful to know how antigen presentation develops in this response rather than just having the day 6 data when mice are near the point of succumbing to cerebral malaria. It would be valuable to know whether DC1 and DC2 present the ETRAMP10.2 antigen early on (eg day 3) and whether this changes over time. This would help align the current study with other work on this topic. In addition, it would be useful to know if naïve DC1 and DC2 can present iRBC in vitro to the BEZ hybridoma.

Referee #2 (Comments on Novelty/Model System):

Novelty is medium - While the CD4 immunopeptidome is highly novel, the techniques are not as well as the finding that cDC1 are important for the development of immune responses to blood stage malaria infections

Medical impact is medium - The results are of fundamental in nature and performed in relevant animal model. The translation of the work remains far off.

Referee #2 (Remarks):

This is a very nice piece of work aimed at characterising the MHC II immunopeptidome presented by dendritic cells during blood stage malaria in mice. The authors also established data on immunodominance hierarchy and showed that CD8+ dendritic cells (cDC1) are superior to other DC subsets for MHC II presentation of malaria-derived peptides. The manuscript needs some grammatical corrections.

Draheim et al

Point-by-point response: our answers are in blue

Referee #1 (Comments on Novelty/Model System):

The science in this report is excellent. The identification of peptide epitopes for MHC II-restricted antigens of *P. berghei* is novel and is the major finding of this report. The effect of depletion of DC1 on the response of CD4 T cells of defined specificity is also novel. The medical impact is distant as it is mainly allowing greater exploration of the mouse model in more detail, for eventual translation. Though, it also provides insight into the role of DC1 in priming of CD4 T cell immunity, which is likely to translate to humans. The *P. berghei* model is an adequate model for examining immunity to malaria, though all animal models have their nuances.

Referee #1 (Remarks):

The report by Draheim and colleagues undertakes a strategy to identify *P. berghei* peptides presented in the context of IAb during blood-stage malaria. It successfully identifies a number of peptide antigens that are able to stimulate CD4 T cells responding to this infection. The authors then generate a hybridoma against one of these peptides and examine which DC subsets are presenting antigen at day 6 of infection. This reveals that cDC1 are the primary presenting population. DC2 are shown to be somewhat defective in antigen presentation at this stage. Depletion of the DC1 population throughout infection reduces the overall CD4 T cell response to *P. berghei* and results in skewing of the remaining response IFN γ /TNF production (Th1) to IL-10 production (Tr1). This suggests DC1 are important in generation of a Th1 biased response in this setting. The experiments are well performed and that findings are valuable and interesting. The identification of MHC II-restricted epitopes will allow greater exploration of this response in future studies.

I have only one additional set of experiments that would enhance this study. It would be useful to know how antigen presentation develops in this response rather than just having the day 6 data when mice are near the point of succumbing to cerebral malaria. It would be valuable to know whether DC1 and DC2 present the ETRAMP10.2 antigen early on (eg day 3) and whether this changes over time. This would help align the current study with other work on this topic. In addition, it would be useful to know if naïve DC1 and DC2 can present iRBC in vitro to the BEZ hybridoma.

We agree that evaluating the kinetics of MHC II presentation by DC1 and DC2 is useful, as it should help discriminate between DC subset-intrinsic differences and specific effects of severe malaria. As suggested, we performed new antigen presentation experiments with DC1 and DC2 isolated from day 3-infected mice and from naïve mice.

The results obtained with naïve DC are presented in three new panels of Fig. 4 (H,I,J). We evaluated the capacity of naïve DC1 and DC2 to present the ETRAMP peptide from pRBC (Fig. 4H) and the OVA peptide from OVA-expressing *E. coli* (Fig. 4I). Like at day 6 post-infection, we found that naïve cDC1 were more potent for processing and presenting pRBC and *E. coli*-derived exogenous antigens on MHC II. We re-assessed the MHC II surface expression by FACS, which allowed us to substantiate our initial analysis (old Fig 4C, new Fig. 4J) and to show that the lower stimulatory activity of DC2 cannot be explained by a reduced surface expression of MHC II. Concerning the presentation of bacterial antigens, a previous study (Schnorrer *et al*, PNAS 2008, PMID 16807294) reported a similar ability of DC1 and DC2 but they used fixed *E. coli* while we fed the DC with live bacteria. Since the origin and folding status of the exogenous antigen are expected to affect processing efficiency, we assume that it could explain this slight discrepancy.

These new results are presented in the results and discussion section (see yellow-highlighted text). Most importantly, they indicate that the defective presentation of ETRAMP by DC2 phagocytosing pRBC may not solely be caused by severe malaria infection. As discussed in the new version of the manuscript, this is in line with a recent Biorxiv-posted study (Fernandez-Ruiz *et al*, <http://biorxiv.org/content/biorxiv/early/2017/03/03/113837.full.pdf>).

Unfortunately, due to the lower sensitivity of our BEZ hybridoma, we did not detect any signal with DC sorted from day 3-infected mice. We are thus unable to show antigen presentation measurements at day 3. Regardless, we think that the data obtained at days 0 and 6 pi are enough to support our conclusion that **cDC1 are superior to cDC2 in MHC II processing and presentation of pRBC-derived antigens and that this difference is not solely due to the infection.**

Furthermore, as an attempt to identify a potential mechanism contributing to the cDC1/cDC2 functional dichotomy in MHC II presentation of pRBC antigens, we assessed the phagocytosis capacity of these 2 subsets in mice infected with GFP-expressing parasites. The results presented in Fig. 4G now suggest that cDC1 display an increased pRBC uptake ability, as compared to cDC2.

Referee #2 (Comments on Novelty/Model System):

Novelty is medium - While the CD4 immunopeptidome is highly novel, the techniques are not as well as the finding that cDC1 are important for the development of immune responses to blood stage malaria infections

Medical impact is medium - The results are of fundamental in nature and performed in relevant animal model. The translation of the work remains far off.

We agree that our work is fundamental in nature. However we think that it provides a first important step in the definition of HLA class II immunopeptidomes for human malaria parasites. Since the mass spectrometry techniques to analyze immunopeptidomes are rapidly evolving and methods have been recently optimized to characterize HLA DR-bound immunopeptidomes with relatively low numbers of cells, we are confident that this is now feasible with *P. falciparum* pRBC. Ultimately, we expect this to improve epitope discovery and help the antigen down-selection process.

As requested by the Editor, we have completed our discussion on translational aspects. The text related to clinical and translational prospects of our study is highlighted in green.

Referee #2 (Remarks):

This is a very nice piece of work aimed at characterising the MHC II immunopeptidome presented by dendritic cells during blood stage malaria in mice. The authors also established data on immunodominance hierarchy and showed that CD8+ dendritic cells (cDC1) are superior to other DC subsets for MHC II presentation of malaria-derived peptides.

The manuscript needs some grammatical corrections..

We agree that English corrections were needed. This has been done in the new version of the manuscript (changes are visible using the track changes feature).

2nd Editorial Decision

16 August 2017

Thank you for the submission of your revised manuscript to EMBO Molecular Medicine. We could unfortunately not get additional input from referee 1 on the revised article, so we asked an expert adviser for help. You will see the comments of this adviser at the end of this email. Please make sure to emphasize the clinical aspect of your study even more than currently by taking into account the comments of our advisor. Please discuss the limitations as mentioned.

In order to accept your manuscript we will also need the following final amendments:

1) Please carefully check the authors guidelines for formatting your Expanded view materials (see: <http://embomolmed.embopress.org/authorguide#expandedview>)

2) In the main manuscript file:

- in legends, provide exact n= and exact p= values, not a range. Please make sure to populate the statistical paragraph according to all the questions asked in the author checklist that you have to fill (see point 3).

Please submit your revised manuscript within two weeks. I look forward to seeing a revised form of your manuscript as soon as possible.

***** Reviewer's comments *****

Adviser:

" [...] I think that there is little translational potential in this manuscript.

Understanding which are the peptides presented and cells responsible to present those peptides is interesting. I would have asked for two more experiments.

On one side I would have asked whether these 3 identified peptides can protect from parasite infection. On the other I would have addressed whether mice lacking DC1 are more susceptible to infection.

This would have provided some more translational aspect."

2nd Revision - authors' response

25 August 2017

Draheim et al – Revision 2

Point-by-point response: our answers are in blue

***** Reviewer's comments *****

Adviser:

" [...] I think that there is little translational potential in this manuscript.

Understanding which are the peptides presented and cells responsible to present those peptides is interesting. I would have asked for two more experiments.

On one side I would have asked whether these 3 identified peptides can protect from parasite infection. On the other I would have addressed whether mice lacking DC1 are more susceptible to infection.

This would have provided some more translational aspect."

These are logical suggestions. However we think that *P. berghei* ANKA infection is not the best model to address these questions and that instead, a self-resolutive model such as *P. chabaudi*, or one that provides long-term protection, such as the genetically attenuated *P. berghei* NK65 ΔHRF, would be better adapted. We have provided some arguments along these lines in the discussion (lines 383-388):

“*PbA* infection is suited to evaluate the role of DC on T cell polarization and pathogenicity during cerebral malaria but this acute model is not well adapted to study the effects of CD4-dependent antibody responses on parasitemia. As previously reported (Piva et al, 2012), we observed no major effect of cDC1 depletion on parasite growth. *Pcc* and *Pb* NK65 ΔHRF are likely to be better models to evaluate the relevance of the identified peptides on parasitemia and antibody-mediated protection.”

Based on PlasmoDB and the Otto *et al* publication (PMID 25359557), half of the identified MHC II peptides are conserved in *Pcc*, 4 display polymorphisms (AMA1, INDK̄NFIATTALSSTEE ; M1, VSEV̄IHPETNYALTG ; END70 KILYNSAKPNSDSH ; PDI LIPEYND̄AAIMLĀEKK) and two are absent (MSP1 and ETRAMP). All peptides are fully conserved in *Pb* NK65.

As a first step to highlight the protective potential of the peptide panel, we analyzed *Pcc*-infected (day 6) and *Pb* NK65 ΔHRF-immunized mice (day 21) and we found that several peptides elicited IFN̄/TNF-production by CD4 T cells. These pilot data have been added as Fig EV2B and EV2C and they are described in the results section (lines 196-206):

“This peptide panel was also relevant in two other malaria models, in which protection strongly relies on humoral responses. In the self-resolutive *Pcc* model, 6 out of 10 peptides tested elicited IFN̄/TNF-producing CD4 responses at day 6 post-infection (Fig. EV2B). Note that the identified ETRAMP and MSP1 peptides are not expressed by *Pcc* due to sequence polymorphisms, hence the absence of reactivity. Another model of interest is the genetically attenuated parasite (GAP) *P.*

berghei NK65 which lacks the Histamine-Releasing Factor (*Pb* NK65 Δ HRF). This vaccine strain, in which the sequences of all identified peptides are conserved (Otto et al, 2014), causes self-resolving blood stage infection and results in long-lasting cross-stage and cross-species immunity (Demarta-Gatsi et al, 2016). Three weeks post-challenge with *Pb* NK65 Δ HRF, we could detect CD4 T cells reactive against 11 of the identified peptides (Fig. EV2C).”

Related to these data, we have added the following statements in the discussion (lines 389-394):

“While our study does not analyze these aspects, we found that some peptides were recognized by CD4 T cells isolated from *Pcc*-infected mice and mice immunized with the *Pb* NK65 Δ HRF GAP, thus setting the stage to assess (i) whether the conserved peptides will elicit protection after challenge with *Pcc* and (ii) whether the protective antibody responses generated by the *Pb* NK65 Δ HRF vaccine GAP target the same MHC II antigens.”

Corresponding Author Name: Nicolas Blanchard
Journal Submitted to: EMBO Molecular Medicine
Manuscript Number: EMM-2017-08123